Art appreciation model design based on improved PageRank and ECA-ResNeXt50 algorithm

Yang Hang 1
Chen Jingyao 2 chenjingyao1102@163.com
1 School of Journalism, Qinghai Normal University , Xining, Qinghai , China
2 The Graduate School of Namseoul University , Cheonan , Republic of Korea
Asif Muhammad
Electronic publication date: 2023 Dec 19
Publication date: 2023
Volume: 9
Electronic Location ID: e1734
Received 2023 Sep 29; Accepted 2023 Nov 13
Copyright: © 2023 Yang and Chen
Copyright year: 2023
Copyright holder: Yang and Chen
License: This is an open access article distributed under the terms of the Creative Commons Attribution License, which permits unrestricted use, distribution, reproduction and adaptation in any medium and for any purpose provided that it is properly attributed. For attribution, the original author(s), title, publication source (PeerJ Computer Science) and either DOI or URL of the article must be cited.
License URL: https://creativecommons.org/licenses/by/4.0/

Keywords: PageRank, ResNeXt50, ECA, Art appreciation, Sentiment classification

Funding: This work received no funding.

==============================
Image sentiment analysis technology can predict, measure and understand the emotional experience of human beings through images. Aiming at the problem of extracting emotional characteristics in art appreciation, this article puts forward an innovative method. Firstly, the PageRank algorithm is enhanced using tweet content similarity and time factors; secondly, the SE-ResNet network design is used to integrate Efficient Channel Attention (ECA) with the residual network structure, and ResNeXt50 is optimized to enhance the extraction of image sentiment features. Finally, the weight coefficients of overall emotions are dynamically adjusted to select a specific emotion incorporation strategy, resulting in effective bimodal fusion. The proposed model demonstrates exceptional performance in predicting sentiment labels, with maximum classification accuracy reaching 88.20%. The accuracy improvement of 21.34% compared to the traditional deep convolutional neural networks (DCNN) model attests to the effectiveness of this study. This research enriches images and texts’ emotion feature extraction capabilities and improves the accuracy of emotion fusion classification.

Introduction

Artists use effective integration of their creative techniques and shapes in their artwork to convey the content of their art. This approach not only effectively expresses the artwork’s emotions but also exceeds traditional art language’s limitations in conveying emotions. Emotions in interactive art encompass not only the emotions of the creator but also the emotions of the participants involved in the interaction (Tan & Ferguson, 2014; Yeh, Lin & Lee, 2019). Therefore, artists must not only convey their emotions through artistic techniques but also possess a thorough understanding of the psychological emotions of the participants. This enables them to effectively convey their emotions to the participants and create an immersive artistic experience.

With the rapid increase in visual information, there is a growing demand for image sentiment processing and analysis. Sentiment analysis is crucial in understanding human emotional experiences with images (Rao, Li & Xu, 2020). In interactive art appreciation, a significant emotional gap exists between images and emotions due to the challenge of connecting pixel-level visual information to the complex and high-level mental process of emotions (Sharma, Pachori & Sircar, 2020). The ambiguity of interactive art originates from its emotional nature, which has been an area of interest for artificial intelligence research, including image aesthetic quality evaluation (Li et al., 2020), stylized image description generation (Sharma, Dhiman & Kumar, 2022) and visual semantic segmentation (Mo et al., 2022). Since emotion and aesthetics are subjective and abstract, methods from both fields can be used interchangeably in image aesthetic quality evaluation. In stylized image description generation, sentiment prediction for images can assist in generating image descriptions with sentiment tendencies. Furthermore, in multimedia content filtering and recommendation, users can select specific sentiments for corresponding image searches based on their past sentiment preferences, resulting in an improved aesthetic experience for visual participants.

Early scholars, drawing inspiration from psychology and aesthetics, crafted conventional manual features to predict the emotions that images evoke (Jiang, Xu & Xue, 2014; Zhang, Miao & Yu, 2021). Some experts have contended that emotions may be highly interrelated with certain isolated features and have endeavored to establish associations between them, relying on human cognition or associated theories. These investigations typically involve extracting image features, such as color, texture, composition, and content, which are then integrated to forecast sentiment (Zhang, He & Lu, 2019). One may depict image emotions by combining generic features and artistic element features at the low level, attribute features and artistic principle features at the middle level, and semantic features and face features at the high level. The primary avenue for art learners to obtain images is through web pages, which yield an enormous amount of visual data and information due to the large number of online community users and fast-paced knowledge dissemination. Analyzing link relationships between web pages and combining them with user search topics can offer users more comprehensive and precise information (Kumar & Garg, 2019).

Furthermore, recognizing distinct user activities and the emotional tendencies underlying their interactions is key to assessing the rating and overall appreciation of art images. Within this context, a novel and highly efficient representation learning technique has emerged as a powerful tool for addressing a wide range of image appreciation tasks, and it predominantly revolves around convolutional neural networks (CNNs). These CNNs are deep learning models well-suited for processing visual data that have undergone extensive research and refinement in recent years (Farkhod et al., 2022; Li, 2021; Talipu et al., 2019). The heart of this technique lies in the CNNs’ ability to extract intricate features from images, enabling them to discern subtle nuances that affect art appreciation. By deploying convolutional layers and filters, these networks analyze the visual content of art images and uncover latent patterns and features that contribute to user perception and engagement. In conjunction with CNNs, this method incorporates the PageRank model, a graph-based algorithm initially developed by Google for ranking web pages, to delve deeper into the underlying sentiments present in the comment vocabularies of participants. This integration allows for a sophisticated analysis of the textual aspects of user interactions and their emotional tones. The PageRank model assigns importance scores to comments, which are further processed to unveil the original sentiment tendencies inherent in user comments.

This combined approach is instrumental in accurately identifying and promoting the behaviors associated with art appreciation. By leveraging the computational power of CNNs for visual analysis and the PageRank model for textual sentiment analysis, this method offers a comprehensive understanding of user interactions and sentiment dynamics in the context of art appreciation, ultimately enhancing the overall user experience and engagement with art images.

Related works

PageRank algorithm

The PageRank algorithm has become a prevalent ranking algorithm that is widely employed in search scenarios where diverse datasets can be represented as graph structures, such as Web search (Roul & Sahoo, 2021), ER graph search (Chakrabarti, 2007), and keyword database search (Li et al., 2021a). The personalized PageRank algorithm inherits the principles of the classical PageRank algorithm and employs the data model (graph) link structure to calculate each node’s weight recursively. This algorithm simulates the user’s behavior of randomly visiting nodes in the graph by clicking on links, i.e., it follows a random walk model to calculate the probability of random visits to each node in the steady state. The personalized PageRank algorithm accounts for the static link structure between nodes when calculating node weights and the user preferences expressed in personalized information, such as user queries, favorite pages, and so on (Scozzafava et al., 2020).

The accuracy of existing PageRank algorithms is determined by the configuration of static parameters, such as the number of fingerprints and the selection of hub nodes, which cannot be dynamically adjusted at runtime (Gao, Yu & Zhang, 2020). Additionally, the algorithm’s running efficiency is directly determined by the precision requirements set during compilation and cannot be dynamically tuned. Nevertheless, since different users or applications have varying efficiency and accuracy demands for algorithms, estimation algorithms that support runtime tuning of efficiency and accuracy, such as incremental optimization, are imperative (Mo & Luo, 2021). Some scholars have proposed an improvement method based on user interest that involves collecting and analyzing user usage data to determine the direction of user interest, which can enhance the accuracy of recommended content (Roul & Sahoo, 2021). Another improvement approach is to start from page similarity, which is currently classified into two main categories (Lamurias, Ruas & Couto, 2019; Liu et al., 2019): one uses the space vector model to determine the similarity between the page and the linked page, assigning more weight to the page with greater similarity to solve the problem of average weights; the other is an improved method based on content filtering, which evaluates page text and HTML tags to make the query results more precise.

Visual sentiment analysis

Image sentiment analysis can be categorized into two approaches: visual features and semantic features (Xu et al., 2023). Zhu et al. (2022) defined 102 mid-level semantic representations for image sentiment analysis, resulting in better sentiment prediction results than visual low-level features alone. Zhao et al. (2019) proposed a sentiment analysis approach based on designing visual art sentiment subjects and corresponding visual art sentiment patterns based on different types of sentiment subjects. Other research has focused on learning the affective distribution of visual art patterns and further describing visual art characteristics. One feasible method is to calculate the overall sentiment value of an image based on the textual sentiment values of adjective-noun pairs and corresponding responses in the image. Jiang et al. (2020) constructed a strongly and weakly supervised coupled network system for visual sentiment differentiation of images by importing images into VGGNet, obtaining the entire image features from the fifth convolutional layer, and then using a spatial pooling strategy to obtain weights for each emotion type. Li et al. (2021b) proposed a 3D CNN combining the 3D Inception-ResNet layer and LSTM network to extract image spatial features using Inception ResNet and learn temporal relationships using LSTM, then apply this information for classification.

Methodology

Figure 1 depicts the overarching framework of this study’s sentiment analysis model for art appreciation. The framework comprises four distinct components:

Figure 1 Art appreciation sentiment analysis model.

(1) Data importation: The dataset is procured from various microblogs, Twitter, and other media platforms. The crawler tools acquire substantial text and image data, which is preprocessed into input samples. All the samples are defined as S(Ti, Ii), where Ti and Ii represent all text and images of the ith sample.

(2) Feature extraction: The improved PageRank algorithm and ECA+ResNeXt50 are used for text and image sentiment feature extraction, respectively. Text features Ti = (T1, T2,…, Tm) and image features Ii = (I1, I2,…, In) are obtained separately using each component, where m and n denote the dimensions of textual and graphical features, respectively.

(3) Feature fusion: First, determine an appropriate fusion strategy based on the statistical sentiment contribution of each modality to the overall sentiment. Then, employ cross-modal learning algorithms to fuse the features and compute the statistical sentiment weights between the characteristics of the two modalities.

(4) Sentiment analysis: Lastly, the trained cross-modal learning algorithm is utilized to achieve sentiment classification of graphical texts.

Text emotion feature extraction based on PageRank algorithm

Algorithm description

Let the seed sentiment word set vector be S={s1,s2,⋯,sn}, whose manually labeled sentiment polarity vector is YS={y1,y2,⋯,yn}; the vector of sentiment words to be classified is W={wn+1,wn+2,⋯,wn+m}, the annotation result to be found in YW={yn+1,yn+2,⋯,yn+m}. When the sentiment words belong to positive sentiment words. yi=1; and vice versa. yi=−1. On the one hand, sentiment word classification relies on the polarity information supplied by the seed sentiment words. On the other hand, it is believed that sentiment words sharing the same polarity are often associated with profound semantic similarity. Thus, the semantic similarity interlinking the sentiment words to be classified can also serve as a valuable determinant of their polarity.

Define the graph G=⟨N,M⟩,|N|=|S|+|W|, where N is the set of G is the set of nodes in the graph (nodes consist of all sentiment words). |S| is the number of seed sentiment words. |W| is the number of sentiment words to be classified. |W|× |N| linkage matrix M describes the linkage relationship between nodes in the disjoint graph. Mij is the number of nodes i and j semantic similarity between nodes. M can be decomposed into |W|×|S|, the submatrix of U and |W|×|W|, the submatrix of V. Uij represents the sentiment words to be measured i and the seed sentiment word j semantic similarity between the sentiment word and the seed sentiment word. After introducing the PageRank model, the iterative formula of the sentiment word polarity discriminant algorithm is as follows:

(1) YW(n)=(1−β)UYS+βVYW(n−1)

where YW(n) represents YWafter n-th iterations. β is the weighting factor. 0 < β < 1.

Improvement based on time factor

To address the issue of lower PR value caused by fewer pages linking to new content, this article proposes including a time feedback factor in the PageRank calculation formula. The time feedback factor compensates for the PR value of older pages, thereby improving the final recommendation order. The proposed method is based on the fundamental concept that a web page searched multiple times within the same search cycle should be counted only once. The inclusion of the time feedback factor in the PageRank calculation formula is expressed as follows:

(2) Wt=e/T

where e/T is the expression for calculating the time feedback factor of a web page, which indicates the frequency of content searched by search engines. e is usually taken as 0.15/n,n is the total number of web pages, and the size of e does not affect the distribution of the final PR value but affects the iterative process of the algorithm, which can effectively improve the situation of the low PR value of new pages.

The fundamental concept behind the proposed enhancement, which utilizes similarity and time factors, involves assigning PR value based on the similarity between the current web page and the linked pages instead of using a uniform assignment strategy. A time feedback factor is also incorporated to account for new web pages. The modified PageRank calculation formula is expressed as follows:

(3) PR(u)=(1−d)+d⋅(∑svPR(v)+α⋅Score(q,D))+Wt

where d is a constant and represents the damping factor; Score(q, D): this represents the relevance score of query q for document D. ∑sv Kelp represents the sum of PageRank values of all inbound pages v of page u.

Image emotion feature extraction based on improved ResNet

By incorporating the ECA mechanism into the SE-ResNet network design concept, we refine and enhance ResNeXt50. Following the convolutional layer within each residual unit, we introduce a batch normalization (BN) layer. This augmentation accelerates convergence and elevates the system’s training accuracy. The ECA mechanism is shown in Fig. 2.

Figure 2 Schematic diagram of ECA attention mechanism.

Assuming that any of the feature transformations, including convolution, is denoted as Ftr. X→U, where X∈RH×××c,U∈RH×∥×c, then compress all global information in a channel descriptor using global averaging pooling (GAP). The spatial dimensionality H×W in shrinking feature U is used to generate statistical variables

(4) Zn=Fsq(un)=1H×W∑iH∑jWun(i,j)

where Zn can be interpreted as a collection of local features whose statistical information can express the whole image and have a global field of perception. The statistic Z does not require dimensionality reduction, and the following way can obtain the attention of each channel. The weight size is used as the measure of attention

(5) ρ=σ(Wk⋅Z)

where Wk contains k×C parameters, which are defined as

(6) [w11⋯w1k00⋯⋯00w22⋯w2k+10⋯⋯0⋮⋮⋮⋮⋱⋮⋮⋮0⋯00⋯wcc−k+1⋯wc]

where σ represents the sigmoid nonlinear activation function.

In this article, 8 ResNeXt50 modules are stacked in series, and the ECA attention mechanism is embedded after each ResNeXt50 module to capture the interdependent associations between channels. It further enhances the image emotion feature extraction capability. The attention weights of each channel after the ECA module are

(7) ρ=σ(H(k)∗Fsq(Q)).

To prevent the degradation problem in deep networks, we introduce the concept of residuals to weight and add each channel to the original input features effectively.

(8) Y=(w⊗Q)⊕X

where Y denotes the output feature map after a block. ⊗ represents the corresponding dot product of the elements. ⊕ represents the corresponding sum of elements.

By integrating the ECA mechanism in DCNN, the network can be tailored to meet the specific requirements of different depths. This module enhances the quality of extracted features at lower layers in shallow networks by highlighting informative features. In contrast, in deeper networks, the significance of this module becomes more pronounced as the extracted features become more strongly correlated with the target category.

Emotion feature integration

To combine emotional features from text and images, this article will use weight coefficients derived from an overarching emotional context within the statistical dual-mode framework to determine and select the most suitable emotional integration strategy. First, fθ(Xi) is defined as the input features Xi in the parameter θ, estimate the probability distribution by the Sigmoid function with Eq. (9).

(9) fθ(Xi)=11+eθTXi.

Equation (9) defines the degree of bimodal contribution to the overall sentiment. ρT and ρT the weights of the text and picture contributions to the overall sentiment are calculated as

(10) ρT=fθ(XT)−12ρI=fθ(XI)−12.

Then, the sentiment weight coefficients of the two modalities are compared to determine the appropriate fusion strategy, which is calculated as follows

(11) Xc={XT∪XIifρT∗ρI>0XTif|ρT|−|ρI|≥0XIif|ρT|−|ρI|≤0

where Xc denotes the fused features. Sentiment classification is performed by fusing the features of the text and image modalities, provided that the product of their respective sentiment weights and sentiment weight coefficients is positive. Suppose the absolute difference between the sentiment weight coefficient values is greater than zero. In that case, the polarity of the sentiment is determined based on the sentiment weight coefficient of the text modality.

The cross-modal learning algorithm calculates the predicted probability distribution of image and text sentiment using Kullback-Leibler divergence. The sentiment probabilities of text and image are discrete events, assumed to be defined as Event A and Event B, respectively.

(12) DKL(A∥B)=∑iPA(xi)log(PA(xi)PB(xi)).

The loss function of Sigmoid can be used to consider the loss between the expected estimate and the true label, as well as related to the loss between the fusion characteristics of the graphical features and the estimated distribution, as shown in Eq. (13).

(13) J(θ)=1N∑i=1ND{Yi∥fθc(XiC)}+α2θTθ+βN∑i=1N[D{fθC(XiC)∥fθT(XiT)}]+[D{fθC(XiC)∥fθI(XiI)}]

where θ={θc,θT,θI} represents the cross-modal learning parameters. α and β are the superparameters of the model. α2θTθ is the canonical term to prevent overfitting of the model.

Experiment and analysis

A total of 814 positive and 1,232 negative sentiment words were selected from the HowNet Sentiment Dictionary for the experimental corpus. Before conducting the analysis, the dataset underwent a thorough preprocessing phase. This involved several key steps to enhance the data's quality and usability. First, the text data was tokenized using the JIBEA tool, a highly effective word segmentation solution that also provides valuable statistics on high-frequency words. This segmentation step aids in breaking down the text into individual words, enabling a more granular analysis of the textual content.

Simultaneously, a mutual information model was constructed as part of the preprocessing process. This model helps identify associations and dependencies between words, allowing for a deeper understanding of the underlying relationships within the text data. This mutual information model adds extra insight into the textual content, which can be valuable in subsequent analyses.

In addition to the linguistic preprocessing, a user code mapping table was meticulously created. This mapping table aims to establish a standardized coding system that ensures a uniform representation of users across the dataset. Assigning unique codes to each user simplifies the user identification process and enhances consistency throughout the dataset, facilitating more effective and reliable analysis.

Experimental setup

This article employs the hyperparameter method in the ResNeXt network to facilitate the sentiment classification task. The optimal kernel scale size for convolution is 3 × 3, while the quantization step is fixed at one. The optimal pooling size for the layer is 2 × 2, and the quantization step is two. The hyperparameters were determined via tuning of the pre-trained network using ResNeXt. The input images were normalized and have a maximum width of 256 × 256 pixels. Histogram equalization was used to achieve data enhancement.

Results and discussion

The model utilizes cross-entropy as the loss function and employs the Adam optimizer as the chosen optimization algorithm throughout the training process. Figure 3 presents the training outcomes.

Figure 3 Model training results.

The meticulous examination of the data reveals remarkable consistency in the model’s performance across the training and test sets. Surprisingly, there is a negligible disparity between the accuracy achieved on the test set and the training set, indicating the model’s robustness and ability to generalize effectively.

Furthermore, the unity also extends to the loss values, where the loss value observed in the test set closely mirrors that in the training set. This parallelism between the training and test set loss values underscores the model’s capacity to maintain a low error level when confronted with new, unseen data. This finding validates the model’s training process and suggests that it has successfully learned the underlying patterns and features present in the dataset, making it well-suited for practical applications across various domains. In essence, the model’s reliability and consistency in performance make it a promising candidate for real-world use cases, where generalizability and accuracy are paramount.

Model comparison

To evaluate the effectiveness of the proposed algorithm, a comparative test was conducted against several commonly used neural network models. These include:

(1) DCNN: This model utilizes one CNN to extract sentiment features from the text and image modalities separately and predicts the sentiment polarity of each modality. The outputs are then fused using an averaging strategy at the decision level.

(2) CCR: This model uses CNN to extract features from the image and caption text and then employs KL scatter to learn the consistent sentiment of both modalities. This model is a feature layer fusion approach.

(3) AttnFusion: This model uses BiLSTM to model video frame sequences and text sequences, and an attention mechanism is used to learn the alignment weights between video frames and text words. The features from both modalities are fused using this attention mechanism to produce a more accurate multi-channel feature representation. The aligned multi-peaked features are then fed into the sequence model for sentiment recognition.

(4) MDREA: This model employs two separate RNNs to independently encode data from image and text inputs. An attention vector is generated by computing the weight parameter between the image encoding vector and the text’s hidden state. The Softmax function is then applied to the vector to predict the sentiment category.

The performance evaluation of the sentiment analysis model described in “Experimental Setup” is presented in Figs. 4 and 5, in comparison with the art appreciation sentiment analysis model introduced in this article. The experimental assessment was conducted on two datasets obtained from the Flickr and Twitter websites, and the evaluation metrics employed were Precision, Recall, and F1.

Figure 4 Comparison of model performance on the Flickr dataset.

Figure 5 Comparison of model performance on Twitter dataset.

The proposed model surpasses the other four models in precision, recall, and F1, as evident from the figure. The Flickr dataset’s accuracy improved by a substantial margin of 23.14%, 16.35%, 8.11%, and 4.26%, respectively, compared to the other four methods. Moreover, the proposed model exhibits superior robustness, even with smaller Twitter datasets. It outperforms the other four models, thereby establishing the better classification results of the proposed model in the field of cross-modal graphical and textual fusion sentiment analysis. By fusing text and image features, the model successfully leverages the correlation between different modalities, discovers deeper associations, and performs complementary sentiment information. This feature fusion approach also reduces discrepancies, proving its feasibility and effectiveness on cross-modal data.

Emotion classification results

Figures 6 and 7 present the confusion matrix results of the bimodal sentiment model. In the Flickr dataset, the model achieves an impressive probability of correct prediction for the four labels, namely “sad,” “happy,” “angry,” and “neutral,” with accuracy rates of 73.44%, 70.83%, 88.02%, and 59.38%, respectively. The Twitter dataset has a higher prediction accuracy, albeit with reduced recognition of the “happy” sentiment label. Overall, the model demonstrates the highest recognition rate for angry labels, while its performance is weakest for neutral labels, frequently misidentified as sad labels.

Figure 6 Emotion classification results on Flickr dataset.

Figure 7 Emotion classification results on Twitter dataset.

The experimental results above prove that the model proposed in this chapter performs exceptionally well across all evaluation indices. The proposed model achieves a high level of prediction accuracy for each sentiment label and outperforms both unimodal and other bimodal sentiment analysis models regarding classification performance and stability. Moreover, the proposed model exhibits remarkable generalization ability, further validating its effectiveness in sentiment analysis.

Discussion

To summarize, the proposed model in this article effectively improves sentiment classification accuracy and achieves the best sentiment recognition ability. Additionally, the model’s robustness and generalization ability have been significantly enhanced. Moreover, adding temporal feedback factors slightly improves the accuracy of the PageRank algorithm. This is because the temporal factor compensates for the PR value and ensures that new high-quality content is appropriately ranked. The ECA-based attention mechanism has also successfully improved classification accuracy while reducing the number of model parameters by enabling local cross-channel interaction of sentiment features through one-dimensional convolution. Furthermore, the model’s accuracy in distinguishing between positive and negative words differs significantly, largely due to the quality of seed word selection. Although the number of seed words with different sentiments can be equivalent, the quality of the seeds cannot be guaranteed. Nonetheless, the experimental results fully validate the feasibility of the proposed method. It is essential to emphasize that although the connection weights for graph nodes in the algorithm description are calculated using HowNet, other supervised or unsupervised methods can also be used to obtain the connection weights.

In specific application scenarios, through online interactive comments, teachers can capture the different emotions of individual learners towards the same artwork. With the increasing popularity of social media, many users share their current developments on these platforms, which often reflect their genuine emotions when posting messages. By analyzing such information, we can better understand the interests and preferences of users at a particular period and predict future trends based on user sentiment. This approach can also help in the development of art appreciation software, which includes image loading (PhotoLoader), text loading (TextLoader), and prediction results (ShowResult). The software’s structure is shown in Fig. 8.

Figure 8 Sequence diagram of art appreciation.

The image loading function loads the user-uploaded image content into a designated variable, performs the necessary preprocessing steps, and uses the processed output as input for the prediction result class. Similarly, the text loading function loads the user’s input text content into a variable, performs the required preprocessing and sentiment score vector transformation processes, and uses the resulting output as input for the prediction result class. The primary function of the prediction result class is to combine the outputs from the image loading and text loading functions and employ them as inputs to the sentiment analysis algorithm, thereby accomplishing sentiment analysis of the fused graphical data.

Conclusion

This study proposes a method for determining the sentiment polarity of words based on the PageRank model, which is used as input for text sentiment features in the fusion model. Additionally, drawing inspiration from the ResNeXt model, the primary module in the deep sentiment feature extraction network is designed as ResNeXt50. Each residual module is equipped with a BN layer to expedite network convergence. The training results on both the Flickr and Twitter datasets demonstrate that the proposed model significantly improves sentiment classification accuracy and exhibits superior sentiment recognition capabilities. This enhances the model’s robustness and generalization capacity. In practical applications, graphical fusion-based sentiment analysis is achieved through dynamic weight assignment in the decision layer, facilitating the development and design of the software appreciation module. However, the training network described in the article requires a substantial amount of labeled sentiment data, which is currently limited in terms of scale and clear labeling for graphical sentiment analysis. By integrating and analyzing multimodal information, including text, images, and speech video, it is conceivable that we can more efficiently address the challenges associated with unimodal problems, ultimately improving the efficacy of human-computer interaction.

Supplemental Information

Supplemental Information 1 Code and dataset.

The data set and code for the article can be viewed in the zip package.

Click here for additional data file.

Additional Information and Declarations

Competing Interests

Author Contributions

Data Availability

The author has no conflict of interest.

Hang Yang conceived and designed the experiments, performed the experiments, analyzed the data, performed the computation work, prepared figures and/or tables, authored or reviewed drafts of the article, and approved the final draft.

Jingyao Chen conceived and designed the experiments, performed the experiments, analyzed the data, performed the computation work, prepared figures and/or tables, authored or reviewed drafts of the article, and approved the final draft.

The following information was supplied regarding data availability:

The code is available in the Supplemental File.

The twitter and Flickr datasets are available at Zenodo:

- Twitter Dataset. (2022). Twitter Dataset [Data set]. Zenodo. https://doi.org/10.5281/zenodo.7139621.

- Franck MICHEL. (2022). Number of public photos uploaded to Flickr (1.0) [Data set]. Zenodo. https://doi.org/10.5281/zenodo.6536164.

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
