# Peer review of "Art appreciation model design based on improved PageRank and ECA-ResNeXt50 algorithm"

_PeerJ Computer Science, doi:10.7717/peerj-cs.1734_

## Round 0.1 · original submission · Major Revisions

Dear Authors,

Thank you for your submission to our journal. After careful consideration and the opinion of the experts in the domain, we feel that your paper has scientific metrics regarding novelty and improvement with respect to traditional Deep Convolutional Neural Networks approaches.

However, the experts believe that the paper should be revised in a couple of aspects, as mentioned in their comments. I have also following suggestions for improvements.

Some re-writing is needed in the abstract eg. the "Deep Convolutional Neural Networks (DCNN) model attests to the effectiveness of this study". needs your attention.

Section 2 could be renamed as Literature Review and include some latest literature as well

Figure 4 and 5, should have Y-Axis labeled started at 0.0 and also give the reference of the studies which has used these DCNN, CCN and other models

**Language Note:** PeerJ staff have identified that the English language needs to be improved. When you prepare your next revision, please either (i) have a colleague who is proficient in English and familiar with the subject matter review your manuscript, or (ii) contact a professional editing service to review your manuscript. PeerJ can provide language editing services - you can contact us at copyediting@peerj.com for pricing (be sure to provide your manuscript number and title). – PeerJ Staff

Reviewer 1 ·

Basic reporting

1. I would suggest that the author change the title to highlight the algorithm model;
2. The description logic of the introduction is not clear enough, I would suggest that the author describe the model in the order proposed;
3. A paragraph should be added at the beginning of each section to explain the section so that the reader understands it.

Experimental design

4. In the methodology section, some formulas lack the corresponding parameter description and function of formulas;
5. What specific improvements were made to ResNet in Section 3.2?
6. How exactly is the data preprocessing in section 4.1 implemented?
7. The explanation of Figure 3 is too little, and the author needs to add a bit more;

Validity of the findings

8. I would suggest the author to add ablation experiments in Part 4 to prove the validity of the model;
9. The references need to select some good articles from the latest journals.

Additional comments

Aiming at the problem of emotional feature extraction in art appreciation, this paper proposes an innovative method. Firstly, PageRank algorithm is improved by using Twitter's content similarity and time factor. Secondly, the SE ResNet network design is used to integrate the high efficiency channel attention (ECA) with the residual network structure. Finally, the weight coefficient of the whole emotion is dynamically adjusted to select a specific emotion fusion strategy, so as to produce an effective bimodal fusion. But there are some shortcomings in this paper:

Reviewer 2 ·

Basic reporting

In this study, the authors have proposed a PageRank model-based method to determine the effective polarity of words, which is used as input to the affective features of the text in the fusion model. In addition, inspired by the ResNeXt model, the main module of the deep emotion feature extraction network is designed as ResNeXt50, and each residual module is equipped with a BN layer to speed up network convergence. The training results of the model on Flickr and Twitter datasets show that the proposed model significantly improves the classification accuracy of emotions and achieves excellent emotion recognition ability, thus enhancing the robustness and generalization ability of the model. However, the following deficiencies need to be improved to complete this paper:

Experimental design

1) The preceding paragraph of the abstract seems to lack a background description;
2) At the end of the introduction section, the author should click to add contributions to this article;
3) The algorithm description in section 3.1 May be considered in tabular form;
4) How are the emotional features integrated in section 3.3?
5) According to the experimental data in section 4.1, the number of emotion words in CNKI is too small to ensure the generalization ability of the experiment;

Validity of the findings

6) In section 4.2, the author needs to give a brief introduction to the evaluation model;
7) Whether the use of different emotion words to carry out emotion analysis in the experiment will cause the model to have no generalization ability;
8) Some contents of the discussion are too detailed, and the author is suggested to delete them appropriately;
9) The conclusion section only summarizes the basic content of this paper but lacks the prospect of the future.

---

## Round 0.2 · accepted · Accept

Dear Authors,
Thank you for improving the paper because of the reviewer's comments; I am happy to let you know that your paper is being recommended for publication. Thank you for your fine contribution.

Reviewer 1 ·

Basic reporting

no comment

Experimental design

no comment

Validity of the findings

no comment

Additional comments

paper is well revised and updated according to the comments, so acceptance is recommended.

Reviewer 2 ·

Basic reporting

All the concerns have been addressed. The paper can be accepted in its current state.

Experimental design

NA

Validity of the findings

NA